# A Particle Swarm Algorithm Based on a Multi-Stage Search Strategy

**DOI:** 10.3390/e23091200

**Published:** 2021-09-11

**Authors:** Yong Shen, Wangzhen Cai, Hongwei Kang, Xingping Sun, Qingyi Chen, Haigang Zhang

**Affiliations:** School of Software, Yunnan University, Kunming 650504, China; sheny@ynu.edu.cn (Y.S.); caiwangzhen1@mail.ynu.edu.cn (W.C.); sunxp@ynu.edu.cn (X.S.); devas9@ynu.edu.cn (Q.C.); portzhang@mail.ynu.edu.cn (H.Z.)

**Keywords:** particle swarm optimization, entropy, strategy, exploration and exploitation, repulsion and attraction

## Abstract

Particle swarm optimization (PSO) has the disadvantages of easily getting trapped in local optima and a low search accuracy. Scores of approaches have been used to improve the diversity, search accuracy, and results of PSO, but the balance between exploration and exploitation remains sub-optimal. Many scholars have divided the population into multiple sub-populations with the aim of managing it in space. In this paper, a multi-stage search strategy that is dominated by mutual repulsion among particles and supplemented by attraction was proposed to control the traits of the population. From the angle of iteration time, the algorithm was able to adequately enhance the entropy of the population under the premise of satisfying the convergence, creating a more balanced search process. The study acquired satisfactory results from the CEC2017 test function by improving the standard PSO and improved PSO.

## 1. Introduction

Scholars have applied different approaches to the increasing number of structurally complex optimization problems, which are difficult to solve using traditional means, including evolutionary algorithms such as the genetic algorithm (GA) [1], bee colony (ABC) algorithm [2], difference (DE) algorithm [3], simulated annealing (SA) [4], ant colony (ACO) algorithm [5], and PSO [6].

PSO was proposed by Kennedy and Eberhart [7] in 1995 as a population-based heuristic optimization algorithm. With the advantages of a simple implementation, high efficiency, and few parameters, it is widely used in fields such as path planning [8], image segmentation [9], neural networks [10,11], data prediction [12], and noise control [13]. However, PSO is prone to getting trapped in local optima and lacks search accuracy in late iterations. To address these issues, PSO has been improved from four main points of view.

(1) Parameter tuning. Gunasundari et al. [14] proposed velocity-bounded Boolean PSO (VbBoPSO) based on binary PSO (BPSO), in which particles are initialized with random binary positions and velocities, with the velocity constrained to a specific range of values [15], to explore more regions and obtain better convergence. Sen et al. [16] proposed adaptive modified particle velocity PSO (MPV-PSO). A method to balance the search capability of PSO was proposed, using a strategy of linearly reducing the inertia weights [17]. Ratnaweera et al. [18] proposed a self-organizing hierarchical particle swarm optimizer with time-varying acceleration coefficients (HPSO-TVAC), which varies the acceleration coefficients to improve the search capabilities. Zhan et al. [19] proposed adaptive particle swarm optimization (APSO) with adaptive control parameters. Chen et al. [20] proposed chaotic dynamic weight particle swarm optimization (CDW-PSO), using inertia quantities with chaotic mapping to modify the search direction.

(2) Learning strategy. PSO is prone to getting trapped in local optima, and the search accuracy is not sufficient in late iterations because it uses only two experiences to guide particle learning. Improving this aspect of the learning strategy has attracted much attention. Liang et al. [21] proposed a dynamic multi-swarm particle swarm optimizer (DMS-PSO) with a dynamic neighborhood structure, in which the learning of each particle is no longer restricted to one population. Liu et al. [22] proposed a hierarchical simple time hierarchy strategy (THSPSO) algorithm using different learning strategies in different search phases. Zhan et al. [23] proposed orthogonal learning particle swarm optimization (OLPSO) with orthogonal learning, in which each particle obtains useful information from its own historical best experience and that of its neighbors. Xu et al. [24] proposed two-swarm learning particle swarm optimization (TSLPSO) based on dimensional learning, which constructs a learning paradigm for each particle by learning each dimension of its individual optimal position from the corresponding dimension of the population optimal position. Finally, Li et al. [25] proposed a learning strategy based on the collaboration of multiple populations to achieve information sharing and co-evolution among populations.

(3) Topology. Kennedy [26] pointed out that the use of topology is effective for population-based algorithms. In structured populations, information is often exchanged between closely linked individuals based on fitness and topological relationships as a way to slow down convergence. Mendes et al. [27] proposed fully informed particle swarm optimization (FIPSO), which uses a fully informed strategy in which each particle is updated based on the historical best experience of its neighbors. Janson et al. [28] constructed a dynamically changing tree topology in which each particle learns from its parent, effectively using the information of each particle.

(4) Algorithm crossover. Hybrid algorithms are a key research area to improve the performance of PSO algorithms. They incorporate operators such as crossover, selection, mutation, and choice to improve the search quality of the population individuals and the general efficiency of the algorithm. For example, the use of genetic operators can improve population diversity and convergence to the global optimum. Hybrid algorithms can better escape local optima and overcome certain inherent drawbacks associated with single algorithms. Zhang et al. [29] proposed differential mutation and novel social learning PSO (DSPSO), which combined four differential variation operations with social learning particle swarm optimization (SLPSO). Nasiraghdam et al. [30] proposed a new approach based on a hybrid genetic algorithm and PSO. Related studies [31,32,33,34] showed that a hybrid PSO algorithm incorporating other evolutionary algorithms not only improves population diversity but prevents premature convergence and increases the probability of finding a global optimal solution.

In summary, the key to PSO improvement lies in balancing diversity and convergence, preventing premature convergence to a local optimum, and improving local exploitation. Previous work used entropy as an evaluation index to improve the algorithm but stopped at using entropy to constrain the population traits and did not explore a more optimal search process. Others divided up the search strategy using both new strategies or vastly different ones. This paper proposes a multi-stage search strategy to fully maintain the population diversity for the global search in the early stage and modifies the update formula in the late stage to help jump out of the local optima, improving the local search ability. While improving the algorithm exploitation ability and search accuracy, the original algorithm converges after adding the strategy. This strategy can be used as an improved operator and implemented in heterogeneous comprehensive learning particle swarm optimization (HCLPSO) and TSLPSO. Both of these algorithms divide sub-populations to balance exploration and exploitation; however, adding a new balancing strategy can have adverse effects. This paper explores these two representative algorithms, which are compatible with the new strategy, to illustrate the effectiveness of the strategy. The experimental results show that this strategy can effectively improve the probability of the algorithm converging to the global optimal solution.

## 2. Related Knowledge

### 2.1. Particle Swarm Optimization

PSO tends to a global convergence through the cooperation and competition of particles in the search space. The particle velocity and position updates are determined by the best position found by the corresponding particle (pbest) and the best position found by the whole population up to the current iteration of the algorithm (gbest). Let the total number of particles be N, the dimension D, and the maximum number of iterations maxiter. Then, the velocity of the *i*th particle at moment t is $v_{i}(t)={\left[v_{i1}(t),v_{i2}(t),\ldots ,v_{im}(t)\right]}^{T}$, the position is $x_{i}(t)={\left[x_{i1}(t),x_{i2}(t),\ldots ,x_{im}(t)\right]}^{T}$, the best position found by the corresponding particle at moment t is $pbest_{i}(t)={\left[p_{i1}(t),p_{i2}(t),\ldots ,p_{im}(t)\right]}^{T}$, and the best position found by the whole population is $gbest={\left[g_{1},g_{2},\ldots ,g_{m}\right]}^{T}$. The particles are iteratively searched according to the following formula: (1)$$v_{i+1}(t+1)=\omega v_{i}(t)+c_{1}r_{1}(pbest_{i}(t)\;-\;x_{i}(t))+c_{2}r_{2}(gbest\;-\;x_{i}(t)),$$
(2)$$x_{i+1}(t+1)=x_{i}(t)+v_{i+1}(t+1),$$
where the inertia weight coefficients are the learning factors. r_1_ and r_2_ are random numbers between [0,1]. The two factors c_1_ and c_2_, known as “acceleration coefficients”, are positive constants commonly used to determine when the cognition speed of the *i*th particle is accelerated towards pbest and gbest, respectively. The symbol ω denotes the inertia weight parameter which was originally developed to address the velocity explosion problem.

### 2.2. TSLPSO Algorithm

Currently, TSLPSO is one of the algorithms that achieves superior results on the congress on evolutionary computation (CEC) test function [24], in which one sub-population uses the learning paradigm constructed by the dimensional learning strategy to guide the local search of particles, and the other uses the learning paradigm constructed by the integrated learning strategy to guide the global search. The learning paradigm constructed by dimensional learning can guide the particles to search in better regions; however, if the majority of particles are near a local optimum and are trapped there, then premature convergence occurs. To solve this problem, the integrated learning strategy is introduced to enhance population diversity and help particles escape local optima.

### 2.3. HCLPSO Algorithm

In HCLPSO, the population is divided into two sub-populations. The first aims to enhance exploration and the second to enhance exploitation. In both sub-populations, samples are generated using an integrated learning (CL) strategy with learning probability Pc to generate random numbers for each dimension of the particles and to compare them with their respective learning probability Pc_i_ values. The velocity of the exploration sub-population is updated as
(3)$$V_{i}^{d}=\omega V_{i}^{d}+c*rand_{i}^{d}*\left(pbest_{f_{i}\left(d\right)}^{d}-X_{i}^{d}\right),$$
and the velocity of the exploitation sub-population as
(4)$$V_{i}^{d}=\omega V_{i}^{d}+c_{1}*rand1_{i}^{d}*\left(pbest_{f_{i}\left(d\right)}^{d}-X_{i}^{d}\right)+c_{2}*rand2_{i}^{d}*\left(gbest^{d}-X_{i}^{d}\right).$$
where $f_{i}\left(d\right)=\left[f_{i}\left(1\right),f_{i}\left(2\right),\ldots ,f_{i}\left(D\right)\right]$ indicates whether the i-th particle of each dimension d follows its own or another’s pbest. When $rand_{i}^{d}$ > Pc_i_, $f_{i}\left(d\right)$ = 1; when $rand_{i}^{d}$ < Pc_i_, then two particles, namely $X_{j}$ and $X_{k}$, are randomly selected from the sub-population of the *i*th particle. If fitness ($X_{j}$) ≤ fitness ($X_{k}$), then $f_{i}\left(d\right)=j$; otherwise, $f_{i}\left(d\right)=k$. 

Since the exploring particles are not allowed to access the information of the exploiting particles, there is no information flow from the exploiting sub-population to the exploring sub-population. To prevent loss of diversity, exploring and exploiting individuals do not interact. Inertia weights and dynamic acceleration coefficients are used for both sub-populations.

## 3. Particle Swarm Optimization with a Multi-Stage Search Strategy

### 3.1. Multi-Stage Search Strategy

The paper uses the multi-stage search strategy to improve the search results based on the original mechanisms of PSO, TSLPSO, and HCLPSO. The multi-stage search strategy is described below.

Before population iteration, each dimension of each particle is given an additional attribute Ra, the value of which is 1 or −1. As shown in Figure 1, blue signifies that the Ra attribute value of the particle is 1, and red signifies that the value of the particle is −1. In subsequent action behavior judgments, this value determines whether two particles are mutually attracted or repelled, i.e., particles attract each other with the opposite Ra value, but repel each other if they are the same. In the first stage, a certain proportion of better adapted particles are selected, and these will not be attracted or repelled, as indicated by the black box at the edge of the particles in Figure 1. The remaining particles will be affected by the better adapted particle swarm, except for those that change their position according to the original algorithm, i.e., those without the black border in the figure, and the strength of the influence is determined by the parameter pow. To prevent the phenomenon of premature clustering, the Euclidean distance between particles is calculated as
(5)$$d\left(X_{i},X_{j}\right)=\sqrt{\sum_{D=1}^{n}{\left(X_{i}^{d}-X_{j}^{d}\right)}^{2}}.$$
when the distance is less than the threshold, the particles with poor fitness will be bounced away.

In the second stage, no operation is applied to the particle population. In the third stage, only the Ra of the optimal particle is kept unchanged, and all other particles have the opposite property value of the optimal particle in each dimension, i.e., they will be attracted by it.

In the velocity update Formula (1), particles are pulled by the gbest and pbest points. This seems similar to the attraction and repulsion of this strategy, but in fact, the PSO easily traps in local optima because particles are influenced by the current gbest point in the early stage, and the poor search accuracy in the later stage is caused by the attraction of the pbest point. Existing improvement algorithms make dynamic linear changes to learning factors c_1_ and c_2_ in the velocity update Formula (1) to achieve the expected improvement. However, from another perspective, the improvement strategy is independent of the velocity and position update Formulate (1) and (2), and this study sought to improve the algorithm in terms of exploration and exploitation and to control the shape of the population by relatively direct and simple means. From the analysis of the experimental results in Section 6, it can be seen that the algorithm’s performance has a potential point of balance between exploration and exploitation, and thus a higher probability of obtaining the optimal solution.

### 3.2. Definition of Particle Actions

#### 3.2.1. Particle Action Definition for Stage 1

The position and velocity of all particles are updated according to velocity and position update Formulae (1) and (2), the position of the lagging particles will change according to the fitness value of the particles, and the direction of movement is determined by the Ra value. The Ra value is an inherent property of each particle: its positive or negative value determines repulsion or attraction between particles, i.e., particles attract those with the opposite Ra value, but repel those with the same. Each particle has a Ra value for each dimension.
(6)$$Ra_{i}^{d}=\{\begin{array}{c} -1\;\;\;\;\;\;rand\left(0,1\right)>0.5\\ 1\;\;\;\;\;\;\;\;\;\;\;\;\;\;\;\;\;\;\;\;\;\;\;\;\;\;\;\;\;\;\;else \end{array}$$

As a result of the randomness of Ra, a lag particle action also has randomness. When the force value pow is large enough, the new interaction force between particles can dominate the movement of the lag particle swarm. The force value is calculated as
(7)$$pow=-\sum_{j=1}^{M}\sum_{d=1}^{D}\left(1-\frac{\left|X_{j}^{d}-X_{i}^{d}\right|}{searchrange}\right)*\frac{Ra_{j}^{d}}{Ra_{i}^{d}}*str,$$
where str is a constant parameter that determines pow, which changes the speed of the particle; the “searchrange” is limited by $X_{min}\;and\;X_{max}.$ The particle itself has a speed, which means that these particles may be pushed away from the gbest point, may be close to the gbest point, or may oscillate in its vicinity. However, since the size of the lag particle population is fixed, only a certain number of particles will act to make up for the difference. If a lag particle improves its fitness beyond that of other particles after iteration, then other particles will replace it as the lag particle population. Put differently, this mechanism ensures that part of the particle swarm is exploited locally, and the rest is explored globally; this study sought a balance between them.

To limit premature convergence, which is the tendency to trap in local optima, the Euclidean distance between particles is incorporated in stage 1. Once the distance between any two particles reaches a threshold, the particle with the lag fitness value will be selected and a force applied to it in the opposite direction with respect to the gbest point, pushing it away from the region in which it was originally located. This strategy coexists with the strategy repulsion and attraction mentioned above, and they affect each other.

#### 3.2.2. Particle Action Definition for Stage 2

As shown in Figure 2, stage 2 is mainly used to judge the population traits and control the transition of the stage; in this study, no additional actions were performed on the particles. By mapping the diversity and convergence of the improvement algorithm, in this study, the parameters were set to control the end of stage 1 and the beginning of stage 3. Stage 2 did not originally exist in our design, but it was found that the experimental results could be improved by incorporating stage 2. Therefore, stage 2 could not be abandoned.

#### 3.2.3. Particle Action Definition for Stage 3

In stage 3, we redefined the additional behavior of the particles, maintaining the Ra value of the current particle closest to the gbest point, and making the Ra values of all other particles the opposite, i.e., all other particles will be attracted by the particle near the gbest point, the power force decreases with the number of iterations, and it will decay to 0 at the end of the iteration.
(8)$$pow=pow-\frac{str*N}{EFS*\left(0.5-stage\right)},$$
where “stage” is the value of the parameter that divides the stage and EFS represents fitness evaluations.

It should be noted that the optimal particle may change, so the pow of each particle will be recalculated after each iteration, and there is no Ra influence between other particles except the optimal particle.

Unlike the standard PSO, in which particles are influenced by the information of the pbest and gbest points, in stage 3, particles are influenced by the information of the current optimal particle position, which does not overlap with the gbest point position. The subtle differences between the two can be seen in Figure 3. The addition of a new guidance factor can help improve the convergence performance of PSO, and such a definition reduces the problem of poor search accuracy caused by the traction of the pbest point in the late stage of the search. Moreover, the current optimal particle position and the gbest point jointly determine the search direction to improve convergence performance.

### 3.3. Particle Property Change History

Ra is randomly attached to each particle at the beginning of phase 1 and will change in the following situations:(1)A particle distance determination is triggered in stage 1. The Ra of the less adapted of two particles will be reassigned with the same value as the other particle;(2)At the beginning of stage 3, a particle that is not the best adapted will have an Ra value opposite to that of the best particle;(3)When the optimal particle changes in stage 3, repeat 2.

### 3.4. Inter-Particle Action

The interactions between particles are summarized as follows, where repulsion and attraction are reflected by the change in the direction of velocity and magnitude, which directly acts on velocity change:(1)In all phases, on the basis of the original algorithm velocity and position update formula, particles that are equivalent to each other will be subject to attraction from the gbest point;(2)In stage 1, the lag particle will be attracted by the pow in Formula (7) from the better particle population;(3)In stage 1, the particle with the most lag of two close particles will be subject to repulsive force;(4)In stage 3, the non-optimal particle will be attracted by the optimal particle.

### 3.5. Framework of RaPSO Algorithm

Input: Tmax is the maximum number of iterations, and Dim is the number of dimensions, Ra: denotes additional properties, and pow denotes force parameters.

Output: GbestValue is the global optimal solution, and a and b are constants for control phase division.

Figure 4 is the framework of the RaPSO algorithm (Algorithm 1).


**Algorithms 1.**
1 Initialize a population, initialize particle velocity, position, fitness; initialize PBest, GBest.2 Randomly assign particles Ra3 for t = 1 to Tmax do4 Update the velocity v of the particles according to the particle population update formula;5      If t < a/Tmax do6      calculate the additional offset of the particle position based on pow and Ra7      update the particle velocity v and position x again.8      update pbest, gbest.9      Distance detection, adjusting Ra.10    else If t > b/Tmax do11        adjust the Ra of the non-optimal particle opposite to the optimal particle.12        Calculate the additional offset of the particle position based on pow and Ra.13        update the particle velocity v and position x again.14    end15 end

## 4. Entropy and Convergence

The paper discusses the effectiveness of the strategy in terms of entropy and convergence. An increase in population confusion often means a decrease in convergence performance, but if it increases the likelihood that the population searches for the optimal value without affecting convergence, i.e., it produces small fluctuations in the convergence curve, it will help improve the algorithm. We chose population entropy and DBscan as reference tools to draw figures to establish that the multi-stage search strategy worked.

### 4.1. Population Entropy

Entropy is a measure of the degree of chaos in a system and population entropy is an indicator of group diversity. The general definition of entropy may not apply in high-dimensional space; therefore, we chose population entropy as the evaluation criterion of algorithm convergence. In this paper, we assume that the number of particles is M, and the best individual position of the particles is f_pb. Calculating population entropy can be divided into three steps:

Step 1: Calculate the minimum fitness of the particle’s historical best position, $f_{\min}=\min\left(f\text{\_}\mathrm{pb}\right)$, and maximum fitness, $f_{\max}=\max\left(f\text{\_}\mathrm{pb}\right)$. The interval of consideration is $\left[f_{\min},f_{\max}\right]$;

Step 2: Divide the interval $\left[f_{\min},f_{\max}\right]$ into M equidistant small regions and calculate the number of f_pb in each small interval, $k_{i},\;i=1,\;2,\;\ldots ,M$, and $\sum_{i=1}^{M}k_{i}=M$;

Step 3: Calculate the population entropy, $E_{t}=-\sum_{i=1}^{M}p_{i}{\mathrm{logp}}_{i},p_{i}=\frac{k_{i}}{M}$.

The larger the computational results of population entropy, the more chaotic the particle distribution in space and the less convergent the algorithm.

We compared PSO and the improved algorithm RaPSO using CEC2017-fun10 as the objective function and by conducting experiments in 10-dimensional space. The fitness value of Figure 5a is higher than that of Figure 5b. As can be seen from Figure 5a, the population entropy value of PSO basically has a continuous decreasing trend, and PSO has nearly no effective exploration in the middle and at the end of the algorithm iteration, after which the fitness value no longer changes. This also demonstrates that PSO is easily trapped in local optima. Figure 5b shows the historical change in the population entropy after adding the stage search strategy, from which we can see that the algorithm is in a more chaotic state from beginning to end, and the population entropy is obviously larger than that of PSO at each moment. However, the experimental results are better than those depicted in Figure 5a, which shows that, under the premise of algorithm convergence, increasing population entropy is beneficial to particles in terms of searching for the optimal solution, i.e., increasing population entropy at the beginning of an iteration can help in global exploration, while increasing it at the end of the iteration can benefit local exploitation.

### 4.2. DBscan

DBscan can measure the degree of population disorder from the perspective of density. It defines clusters as the maximum set of densely connected points, which can divide regions with sufficient density into clusters and find clusters of arbitrary shapes in the spatial database of noise. The DBscan calculation method is described as follows:

Step 1: Define the field radius e and the threshold of core point;

Step 2: Start from one particle and find all particles that have a density connection. Jump out of this cluster and start to find particles again from the other particle;

Step 3: Repeat step 2 until all particles have been traversed.

We chose the conditions for the population entropy experiment and plotted Figure 6. From the figure, we can see that DBscan can depict the clustering phenomenon of the population more graphically than population entropy. The PSO algorithm enters a high degree of clustering after 400 iterations, where the particle population cannot jump out of the local optimum due to insufficient guidance information. Similarly, it can be seen that the overall convergence performance of the population remains unchanged after adding the strategy, while the degree of confusion is improved.

## 5. Experimental Analysis of the Improved Algorithm

### 5.1. Experimental Design Method

We chose 29 test functions in CEC2017 as objective functions for optimization. Among them, Fun1 is single-peaked, Fun3–Fun10 are multi-peaked, Fun11–Fun20 are mixed, and Fun21–Fun30 are composite. The details are shown in the Table 1.

### 5.2. Experimental Parameter Selection

The 29 test functions of CEC2017 were selected as target functions for optimization tests, and experiments were conducted with a population size N = 100, repeat = 30, and D = 10, 30. Other parameter settings are shown in Table 2. In order to achieve a better comparison, the parameters were all selected based on previous experiments [24].

After adding the improvement mechanism, there were three additional controllable parameters in each algorithm: str, prop, and stage. In order to ensure the accuracy of parameter selection, the number of experimental repetitions was increased to 100.

#### 5.2.1. The Parameter Str

The parameter str affects the magnitude of the strength of all repulsions and attractions in stages 1 and 3. The algorithm used in the experiment was the standard PSO. With the other parameters unchanged, Fun5, Fun14, and Fun23 in CEC2017 were selected as objects for the comparison experiments. The *x*-axis in Figure 7 represents the value of parameter str, and the *y*-axis represents the average fitness after normalization. The value range of the *x*-axis was obtained after a large number of experiments. The experimental range was far from that. It was the range obtained after preliminary screening. The sub-graph in the Figure 7 is a detailed display of the *x*-axis from 20 to 30. The results from Figure 7 show that str = 25 was chosen to obtain better results.

#### 5.2.2. The Parameter Prop

The parameter prop determines the proportion of poorly adapted particles in stage 1. Because only Fun5 in CEC2017 was selected as the test function, the fitness value of the *y*-axis was unprocessed. Other parts are similar to those used in the experiment in Section 5.2.1 Based on the experimental results, a value of 0.5 was chosen in Figure 8, meaning that half of the particles were attracted to or repelled by the other half.

#### 5.2.3. The Parameter Stage

The parameter stage is the basis for deciding the division of the three stages. Similarly, we used standard PSO as the experimental algorithm in 10 dimensions. The test function was Fun25 in CEC2017. Based on the experimental results in Table 3, this was set to 0.1, which means that the first stage accounts for 40% of the iterations, the second stage 20%, and the third stage 40%.

### 5.3. Analysis of Experimental Results

#### 5.3.1. Standard PSO Based on Multi-Stage Search Strategy

Table 4 and Table 5 is the experiment data of results of PSO and RaPSO in the case of 10/30 dimensions according to the parameters mentioned in Section 5.2. The data in bold in the tables is preferred.

#### 5.3.2. TSLPSO Based on Multi-Stage Search Strategy

Table 6 and Table 7 is the experiment data of results of TSLPSO and RaTSLPSO in the case of 10/30 dimensions according to the parameters mentioned in Section 5.2. The data in bold in the tables is preferred.

#### 5.3.3. HCLPSO Based on Multi-Stage Search Strategy

Table 8 and Table 9 is the experiment data of results of HCLPSO and RaHCLPSO in the case of 10/30 dimensions according to the parameters mentioned in Section 5.2. The data in bold in the tables is preferred.

## 6. Analysis of the Experimental Results

Figure 9 shows various test functions of more significant optimization effects from the six tables in Section 5.3. The *x*-axis is the fitness evaluation number (Fes, 10 dimensions equal 10^5^; 30 dimensions equal 3 × 10^5^). The *y*-axis is the average fitness value. Different colors represent different original algorithms. Solid lines represent algorithms with the strategy and dotted lines represent those without. According to the figure, algorithms adopting the strategy achieve lower (better) fitness and better search performance. Since the strategy increases population disorder during early iterations, the phenomena of slower convergence in early iterations and a poorer fitness value could be observed in certain circumstances in the figure; however, these phenomena are often improved at the end of search. Full development in the preliminary stage lays a more solid foundation for later exploration, and may in turn yield better fitness values. According to Figure 9, the red dotted curve of PSO is always the first stabilized, which means that PSO is more likely to be trapped in local optima. Both TSLPSO and HCLPSO can improve this. However, the assistance of the new strategy can produce better results. According to the figure, the strategy has a more significant optimizing effect on PSO and its two improved algorithms. The same conclusion is drawn in 10 dimensions and 30 dimensions, which further verifies the effectiveness of the strategy as regards the scope of the application.

## 7. Conclusions

A relatively direct and simple method is proposed herein to address problems such as premature convergence and poor searching precision with PSO, i.e., improving particle swarm diversity in early iterations of the algorithm and introducing guidance for the best particle location in the population at the end of the iteration to enhance searching precision. The concept was based on PSO and its two improved algorithms and acts as a multi-stage search strategy operator. It guides the particle swarm to different goals in different stages, improves the comprehensive exploration and development ability, and upgrades population disorder without changing the convergence performance of the algorithm too much, thus effectively improving the particle’s ability to jump out of local optima. Particle swarm behavior before and after the application of the strategy was compared and described, and the effectiveness of the strategy was verified with the population entropy and DBscan tools. The contrast experiment indicated the selection process for three parameters in the proposed strategy. Moreover, according to experimental results of the algorithms on the test functions, the strategy can improve search performance.

In future work, we hope to mathematically demonstrate that the selected parameter values are optimal, prove the general applicability of the strategy, and demonstrate the adequacy and necessity of the divided stages. The effectiveness of the proposed strategy was experimentally demonstrated, but a mathematical analysis and proof are required.

## Figures and Tables

**Figure 1 entropy-23-01200-f001:**
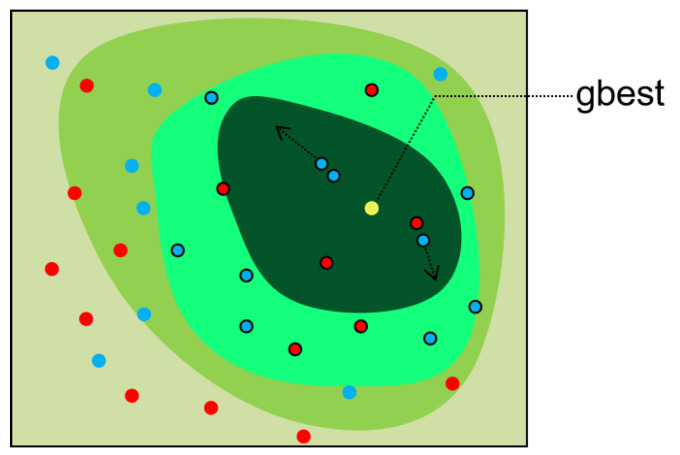
Particle swarm optimization in search strategy 1.

**Figure 2 entropy-23-01200-f002:**
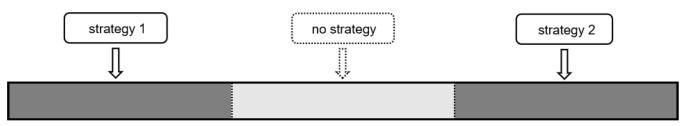
Axis diagram of population iteration number.

**Figure 3 entropy-23-01200-f003:**
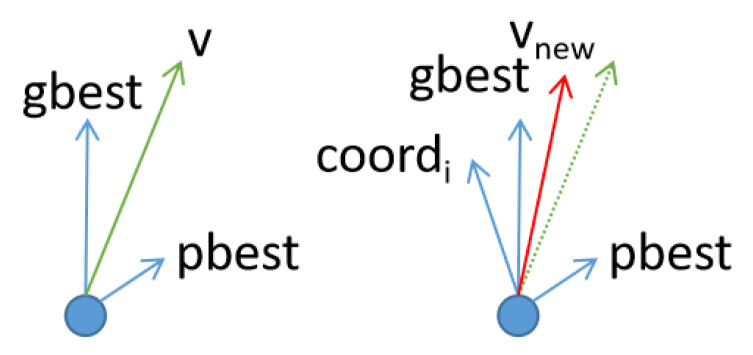
Particle velocity diagram in stage 3.

**Figure 4 entropy-23-01200-f004:**
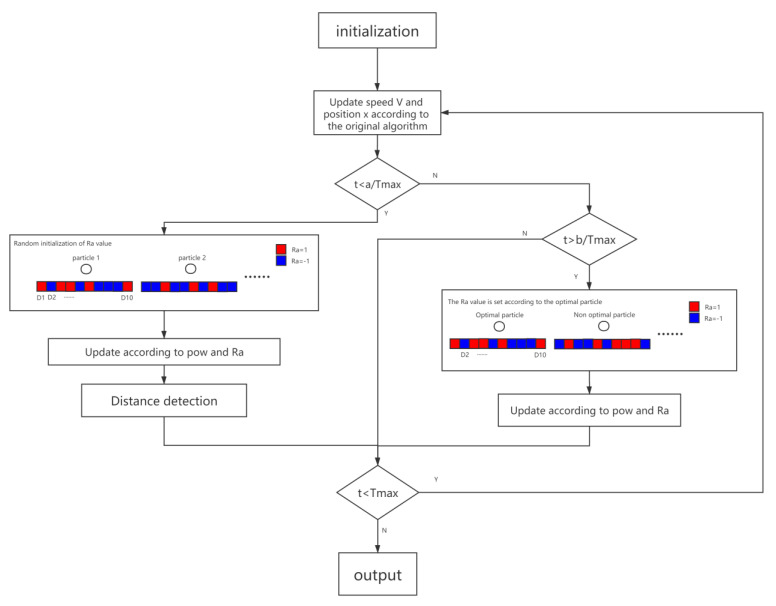
Framework of the RaPSO algorithm.

**Figure 5 entropy-23-01200-f005:**
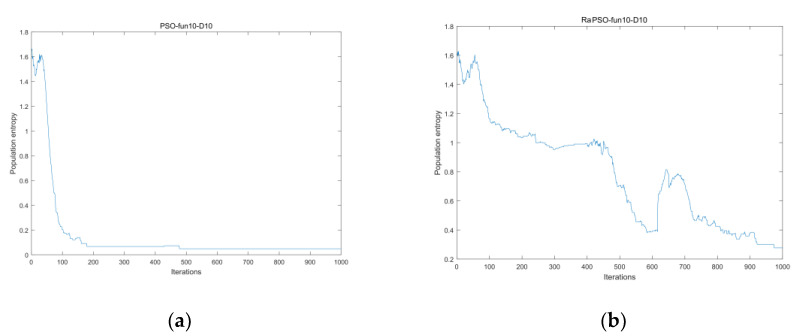
(**a**) Population entropy without strategy; (**b**) population entropy with strategy.

**Figure 6 entropy-23-01200-f006:**
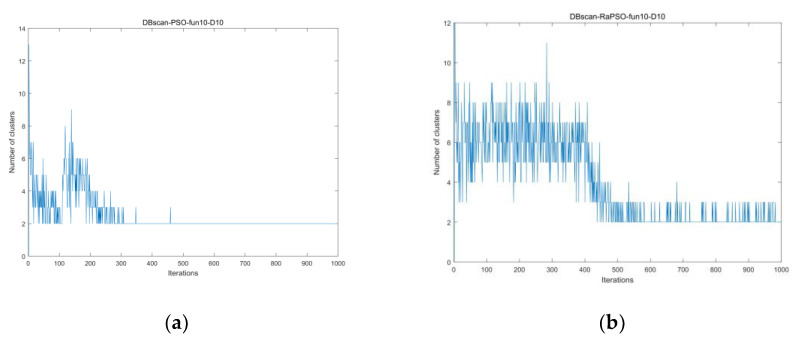
(**a**) The number of clusters without applying policies; (**b**) the number of clusters applying policies.

**Figure 7 entropy-23-01200-f007:**
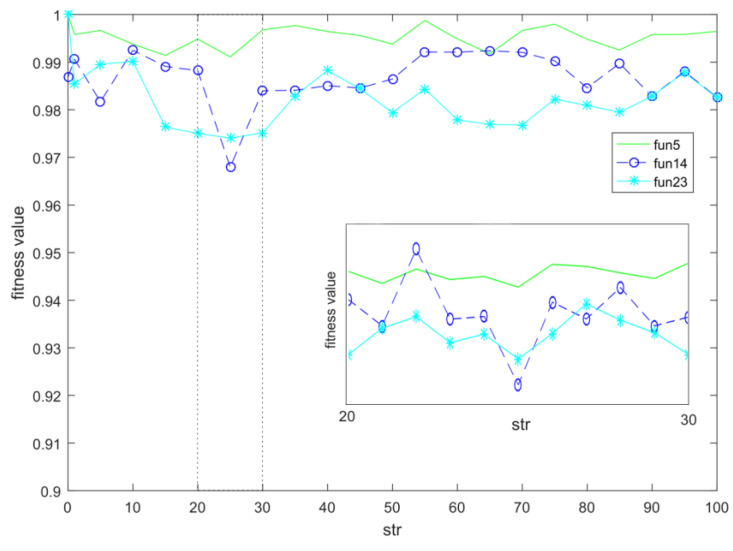
Str parameter selection.

**Figure 8 entropy-23-01200-f008:**
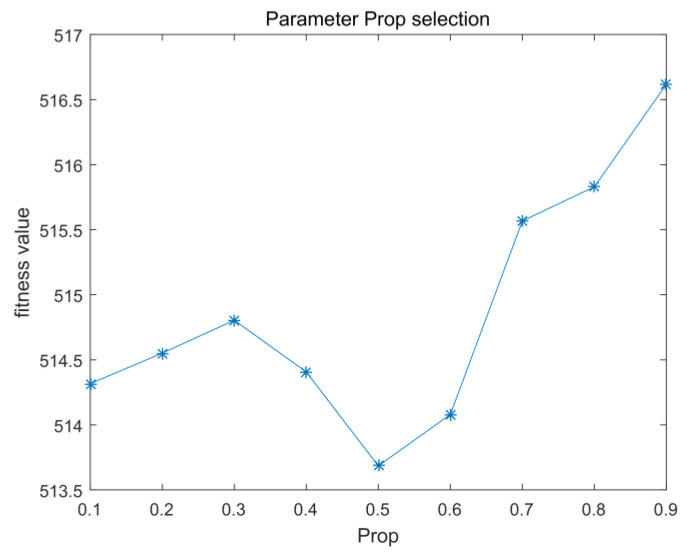
Prop parameter selection.

**Figure 9 entropy-23-01200-f009:**
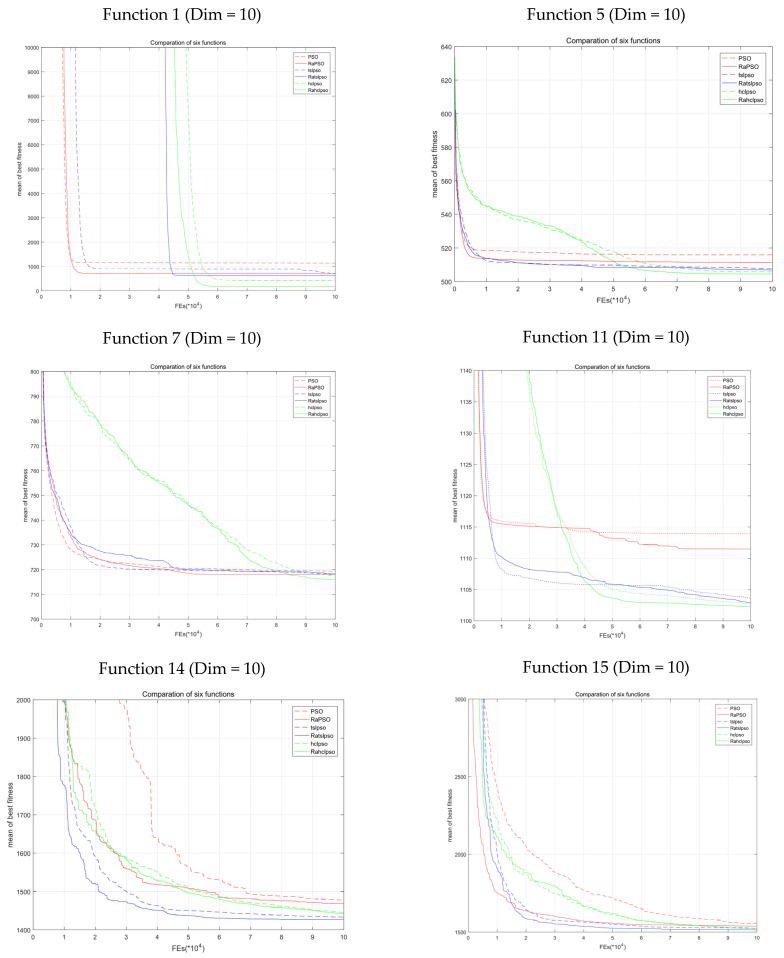
Comparison of the six algorithms.

**Table 1 entropy-23-01200-t001:** CEC17 functions. U: unimodal; M: multimodal; H: hybrid; C: composition.

Num	Function Name	Property	Best Value
Fun1	Shifted and Rotated Bent Cigar Function	U	100
Fun3	Shifted and Rotated Zakharov Function	M	300
Fun4	Shifted and Rotated Rosenbrock’s Function	M	400
Fun5	Shifted and Rotated Rastrigin’s Function	M	500
Fun6	Shifted and Rotated Expanded Scaffer’s F6 Function	M	600
Fun7	Shifted and Rotated Lunacek Bi_Rastrigin Function	M	700
Fun8	Shifted and Rotated Non-Continuous Rastrigin’s Function	M	800
Fun9	Shifted and Rotated Levy Function	M	900
Fun10	Shifted and Rotated Schwefel’s Function	M	1000
Fun11	Hybrid Function 1 (N = 3)	H	1100
Fun12	Hybrid Function 2 (N = 3)	H	1200
Fun13	Hybrid Function 3 (N = 3)	H	1300
Fun14	Hybrid Function 4 (N = 4)	H	1400
Fun15	Hybrid Function 5 (N = 4)	H	1500
Fun16	Hybrid Function 6 (N = 4)	H	1600
Fun17	Hybrid Function 6 (N = 5)	H	1700
Fun18	Hybrid Function 6 (N = 5)	H	1800
Fun19	Hybrid Function 6 (N = 5)	H	1900
Fun20	Hybrid Function 6 (N = 6)	H	2000
Fun21	Composition Function 1 (N = 3)	C	2100
Fun22	Composition Function 2 (N = 3)	C	2200
Fun23	Composition Function 3 (N = 4)	C	2300
Fun24	Composition Function 4 (N = 4)	C	2400
Fun25	Composition Function 5 (N = 5)	C	2500
Fun26	Composition Function 6 (N = 5)	C	2600
Fun27	Composition Function 7 (N = 6)	C	2700
Fun28	Composition Function 8 (N = 6)	C	2800
Fun29	Composition Function 9 (N = 3)	C	2900
Fun30	Composition Function 10 (N = 3)	C	3000

**Table 2 entropy-23-01200-t002:** Parameter selection of the three algorithms.

PSO	$w=0.5,c_{1}=c_{2}=2$
TSLPSO	$w=0.9-0.4,c_{1}=c_{2}=1.49445,c_{3}=0.5-2.5$
HCLPSO	$w=0.99-0.2,c_{1}=2.5-0.5,c_{2}=0.5-2.5,c=3-1.5,g_{1}=37,g_{2}=63$

**Table 3 entropy-23-01200-t003:** Stage parameter selection.

Stage Value	Min	Max	Ave	Std
0	2893.39	2954.96	2931.93	22.69
0.05	2893.58	2956.03	2928.58	23.68
0.1	2892.97	2957.38	**2927.66**	25.22
0.15	2893.77	2957.98	2934.97	22.25
0.2	2892.86	2959.27	2935.50	21.88
0.25	2892.85	2953.28	2934.69	20.67
0.3	2893.24	2960.55	2933.29	22.98
0.35	2893.69	2956.03	2932.67	22.84
0.4	2892.68	2955.04	2931.57	23.60
0.45	2893.62	2959.70	2932.11	22.51
0.5	2893.16	2953.78	2934.50	21.79

**Table 4 entropy-23-01200-t004:** Comparison of experimental results of PSO and RaPSO in the case of 10 dimensions.

Fun	PSO (Dim = 10)				RaPSO (Dim = 10)			
	Min	Max	Mean	Std	Min	Max	Mean	Std
F1	**1.02 × 10^2^**	2.54 × 10^3^	1.29 × 10^3^	8.30 × 10^2^	1.13 × 10^2^	**1.53 × 10^3^**	**7.13 × 10^2^**	**5.06 × 10^2^**
F3	3.00 × 10^2^	3.00 × 10^2^	3.00 × 10^2^	**1.06 × 10^−14^**	3.00 × 10^2^	3.00 × 10^2^	3.00 × 10^2^	9.64 × 10^−11^
F4	**4.00 × 10^2^**	4.35 × 10^2^	4.17 × 10^2^	**1.66 × 10^1^**	4.00 × 10^2^	4.35 × 10^2^	**4.16 × 10^2^**	1.66 × 10^1^
F5	5.04 × 10^2^	5.34 × 10^2^	5.16 × 10^2^	7.92 × 10^0^	5.04 × 10^2^	**5.18 × 10^2^**	**5.11 × 10^2^**	**3.53 × 10^0^**
F6	6.00 × 10^2^	6.00 × 10^2^	6.00 × 10^2^	**4.22 × 10^−14^**	6.00 × 10^2^	6.00 × 10^2^	6.00 × 10^2^	3.69 × 10^−13^
F7	**7.13 × 10^2^**	7.28 × 10^2^	7.20 × 10^2^	3.77 × 10^0^	7.14 × 10^2^	**7.22 × 10^2^**	**7.18 × 10^2^**	**2.29 × 10^0^**
F8	8.10 × 10^2^	8.40 × 10^2^	8.21 × 10^2^	7.84 × 10^0^	**8.06 × 10^2^**	**8.19 × 10^2^**	**8.13 × 10^2^**	**3.22 × 10^0^**
F9	9.00 × 10^2^	9.00 × 10^2^	9.00 × 10^2^	0.00 × 10^0^	9.00 × 10^2^	9.00 × 10^2^	9.00 × 10^2^	0.00 × 10^0^
F10	1.13 × 10^3^	1.85 × 10^3^	1.42 × 10^3^	1.93 × 10^2^	**1.01 × 10^3^**	**1.51 × 10^3^**	**1.30 × 10^3^**	**1.38 × 10^2^**
F11	**1.10 × 10^3^**	1.13 × 10^3^	1.11 × 10^3^	7.65 × 10^0^	1.10 × 10^3^	**1.12 × 10^3^**	**1.11 × 10^3^**	**4.54 × 10^0^**
F12	**1.50 × 10^3^**	4.31 × 10^5^	2.66 × 10^4^	7.69 × 10^4^	1.72 × 10^3^	**2.37 × 10^4^**	**1.13 × 10^4^**	**6.96 × 10^3^**
F13	**1.31 × 10^3^**	8.14 × 10^3^	3.09 × 10^3^	2.00 × 10^3^	1.31 × 10^3^	**4.62 × 10^3^**	**2.33 × 10^3^**	**9.75 × 10^2^**
F14	**1.44 × 10^3^**	1.56 × 10^3^	1.48 × 10^3^	2.63 × 10^1^	1.44 × 10^3^	**1.49 × 10^3^**	**1.47 × 10^3^**	**1.32 × 10^1^**
F15	1.51 × 10^3^	1.73 × 10^3^	1.56 × 10^3^	5.09 × 10^1^	**1.50 × 10^3^**	**1.58 × 10^3^**	**1.54 × 10^3^**	**1.96 × 10^1^**
F16	**1.60 × 10^3^**	1.85 × 10^3^	1.71 × 10^3^	7.88 × 10^1^	1.60 × 10^3^	**1.73 × 10^3^**	**1.69 × 10^3^**	**5.15 × 10^1^**
F17	**1.72 × 10^3^**	1.81 × 10^3^	1.75 × 10^3^	1.94 × 10^1^	1.73 × 10^3^	**1.75 × 10^3^**	**1.74 × 10^3^**	**8.12 × 10^0^**
F18	**1.87 × 10^3^**	2.15 × 10^4^	7.02 × 10^3^	4.88 × 10^3^	2.00 × 10^3^	**1.12 × 10^4^**	**6.13 × 10^3^**	**2.85 × 10^3^**
F19	**1.90 × 10^3^**	2.00 × 10^3^	1.92 × 10^3^	2.22 × 10^1^	1.91 × 10^3^	**1.94 × 10^3^**	**1.92 × 10^3^**	**8.57 × 10^0^**
F20	2.01 × 10^3^	2.20 × 10^3^	2.07 × 10^3^	5.13 × 10^1^	**2.01 × 10^3^**	**2.09 × 10^3^**	**2.04 × 10^3^**	**1.95 × 10^1^**
F21	2.25 × 10^3^	2.25 × 10^3^	2.25 × 10^3^	**8.44 × 10^−14^**	**2.20 × 10^3^**	**2.20 × 10^3^**	**2.20 × 10^3^**	1.19 × 10^−13^
F22	2.21 × 10^3^	2.35 × 10^3^	2.35 × 10^3^	2.56 × 10^1^	**2.20 × 10^3^**	**2.30 × 10^3^**	**2.30 × 10^3^**	**1.83 × 10^1^**
F23	2.40 × 10^3^	2.83 × 10^3^	2.67 × 10^3^	1.18 × 10^2^	2.40 × 10^3^	**2.69 × 10^3^**	**2.63 × 10^3^**	**9.40 × 10^1^**
F24	2.50 × 10^3^	2.83 × 10^3^	2.62 × 10^3^	9.09 × 10^1^	2.50 × 10^3^	**2.60 × 10^3^**	**2.58 × 10^3^**	**3.79 × 10^1^**
F25	**2.89 × 10^3^**	2.96 × 10^3^	**2.92 × 10^3^**	2.55 × 10^1^	2.90 × 10^3^	**2.95 × 10^3^**	2.93 × 10^3^	**2.15 × 10^1^**
F26	2.80 × 10^3^	3.76 × 10^3^	2.94 × 10^3^	2.67 × 10^2^	**2.60 × 10^3^**	**2.90 × 10^3^**	**2.84 × 10^3^**	**8.14 × 10^1^**
F27	3.15 × 10^3^	3.46 × 10^3^	3.27 × 10^3^	8.58 × 10^1^	**3.10 × 10^3^**	**3.28 × 10^3^**	**3.18 × 10^3^**	**5.27 × 10^1^**
F28	3.10 × 10^3^	3.15 × 10^3^	3.14 × 10^3^	**2.21 × 10^1^**	**3.10 × 10^3^**	**3.15 × 10^3^**	**3.13 × 10^3^**	2.35 × 10^1^
F29	3.14 × 10^3^	3.30 × 10^3^	3.18 × 10^3^	3.60 × 10^1^	**3.14 × 10^3^**	**3.20 × 10^3^**	**3.17 × 10^3^**	**1.45 × 10^1^**
F30	3.50 × 10^3^	3.70 × 10^4^	1.15 × 10^4^	9.46 × 10^3^	**3.43 × 10^3^**	**1.04 × 10^4^**	**5.61 × 10^3^**	**1.91 × 10^3^**

**Table 5 entropy-23-01200-t005:** Comparison of experimental results of PSO and RaPSO in the case of 30 dimensions.

Fun	PSO (Dim = 30)				RaPSO (Dim = 30)			
	Min	Max	Mean	Std	Min	Max	Mean	Std
F1	**1.00 × 10^2^**	1.07 × 10^9^	1.02 × 10^8^	3.13 × 10^8^	1.00 × 10^2^	**1.91 × 10^3^**	**2.42 × 10^2^**	**3.72 × 10^2^**
F3	**3.02 × 10^2^**	4.20 × 10^2^	3.27 × 10^2^	2.60 × 10^1^	**4.03 × 10^2^**	5.69 × 10^2^	4.81 × 10^2^	5.06 × 10^1^
F4	4.04 × 10^2^	5.80 × 10^2^	4.86 × 10^2^	4.48 × 10^1^	**4.04 × 10^2^**	**4.96 × 10^2^**	**4.70 × 10^2^**	**2.00 × 10^1^**
F5	5.66 × 10^2^	6.57 × 10^2^	6.05 × 10^2^	2.48 × 10^1^	**5.58 × 10^2^**	**6.04 × 10^2^**	**5.88 × 10^2^**	**1.24 × 10^1^**
F6	6.00 × 10^2^	6.25 × 10^2^	6.07 × 10^2^	6.46 × 10^0^	**6.00 × 10^2^**	**6.08 × 10^2^**	**6.04 × 10^2^**	**2.43 × 10^0^**
F7	7.75 × 10^2^	9.14 × 10^2^	8.21 × 10^2^	2.96 × 10^1^	**7.70 × 10^2^**	**8.29 × 10^2^**	**8.00 × 10^2^**	**1.74 × 10^1^**
F8	8.83 × 10^2^	9.94 × 10^2^	9.21 × 10^2^	3.13 × 10^1^	**8.67 × 10^2^**	**9.34 × 10^2^**	**9.10 × 10^2^**	**1.74 × 10^1^**
F9	1.01 × 10^3^	5.26 × 10^3^	2.83 × 10^3^	1.16 × 10^3^	**9.02 × 10^2^**	**2.81 × 10^3^**	**1.94 × 10^3^**	**6.10 × 10^2^**
F10	2.86 × 10^3^	4.84 × 10^3^	3.89 × 10^3^	5.55 × 10^2^	**2.60 × 10^3^**	**4.47 × 10^3^**	**3.78 × 10^3^**	**5.39 × 10^2^**
F11	1.17 × 10^3^	1.38 × 10^3^	1.24 × 10^3^	4.55 × 10^1^	**1.16 × 10^3^**	**1.28 × 10^3^**	**1.23 × 10^3^**	**3.38 × 10^1^**
F12	3.02 × 10^3^	1.80 × 10^7^	1.16 × 10^6^	4.33 × 10^6^	**2.41 × 10^3^**	**1.51 × 10^4^**	**6.47 × 10^3^**	**3.37 × 10^3^**
F13	**1.33 × 10^3^**	2.36 × 10^4^	2.73 × 10^3^	4.05 × 10^3^	1.36 × 10^3^	**2.62 × 10^3^**	**1.77 × 10^3^**	**3.73 × 10^2^**
F14	**1.52 × 10^3^**	1.96 × 10^3^	1.67 × 10^3^	1.06 × 10^2^	1.56 × 10^3^	**1.76 × 10^3^**	**1.65 × 10^3^**	**6.13 × 10^1^**
F15	**1.53 × 10^3^**	1.98 × 10^3^	1.60 × 10^3^	7.78 × 10^1^	1.54 × 10^3^	**1.62 × 10^3^**	**1.58 × 10^3^**	**2.20 × 10^1^**
F16	2.11 × 10^3^	2.98 × 10^3^	2.54 × 10^3^	2.28 × 10^2^	**1.76 × 10^3^**	**2.40 × 10^3^**	**2.11 × 10^3^**	**1.77 × 10^2^**
F17	**1.78 × 10^3^**	2.69 × 10^3^	2.12 × 10^3^	2.01 × 10^2^	1.79 × 10^3^	**2.09 × 10^3^**	**1.95 × 10^3^**	**8.23 × 10^1^**
F18	**1.01 × 10^4^**	1.38 × 10^5^	5.53 × 10^4^	3.32 × 10^4^	1.12 × 10^4^	**8.45 × 10^4^**	**3.96 × 10^4^**	**1.92 × 10^4^**
F19	2.01 × 10^3^	1.64 × 10^4^	5.69 × 10^3^	3.76 × 10^3^	**1.94 × 10^3^**	**5.62 × 10^3^**	**3.06 × 10^3^**	**1.01 × 10^3^**
F20	**2.26 × 10^3^**	2.74 × 10^3^	2.47 × 10^3^	1.37 × 10^2^	2.26 × 10^3^	**2.56 × 10^3^**	**2.42 × 10^3^**	**7.98 × 10^1^**
F21	**2.20 × 10^3^**	**2.20 × 10^3^**	**2.20 × 10^3^**	**4.78 × 10^−13^**	2.25 × 10^3^	2.25 × 10^3^	2.25 × 10^3^	5.00 × 10^−13^
F22	**2.30 × 10^3^**	**2.30 × 10^3^**	**2.30 × 10^3^**	4.55 × 10^−13^	2.35 × 10^3^	2.35 × 10^3^	2.35 × 10^3^	4.55 × 10^−13^
F23	**2.96 × 10^3^**	4.50 × 10^3^	3.53 × 10^3^	3.57 × 10^2^	2.98 × 10^3^	**4.01 × 10^3^**	**3.46 × 10^3^**	**3.14 × 10^2^**
F24	2.60 × 10^3^	2.61 × 10^3^	2.60 × 10^3^	1.62 × 10^0^	2.60 × 10^3^	**2.60 × 10^3^**	**2.60 × 10^3^**	**1.16 × 10^−12^**
F25	2.90 × 10^3^	3.12 × 10^3^	2.97 × 10^3^	5.84 × 10^1^	**2.90 × 10^3^**	**2.97 × 10^3^**	**2.93 × 10^3^**	**3.08 × 101**
F26	2.80 × 10^3^	5.43 × 10^3^	2.90 × 10^3^	4.79 × 10^2^	2.80 × 10^3^	**2.80 × 10^3^**	**2.80 × 10^3^**	**1.50 × 10^−12^**
F27	3.81 × 10^3^	5.30 × 10^3^	4.53 × 10^3^	3.39 × 10^2^	**3.73 × 10^3^**	**4.47 × 10^3^**	**4.17 × 10^3^**	**2.25 × 10^2^**
F28	3.22 × 10^3^	3.42 × 10^3^	3.30 × 10^3^	5.42 × 10^1^	**3.22 × 10^3^**	**3.30 × 10^3^**	**3.26 × 10^3^**	**2.43 × 10^1^**
F29	3.31 × 10^3^	4.10 × 10^3^	3.69 × 10^3^	2.25 × 10^2^	**3.28 × 10^3^**	**3.63 × 10^3^**	**3.44 × 10^3^**	**8.41 × 10^1^**
F30	4.23 × 10^3^	5.44 × 10^4^	9.02 × 10^3^	9.30 × 10^3^	**4.09 × 10^3^**	**1.43 × 10^4^**	**6.76 × 10^3^**	**2.78 × 10^3^**

**Table 6 entropy-23-01200-t006:** Comparison of experimental results of TSLPSO and RaTSLPSO in the case of 10 dimensions.

Fun	TSLPSO (Dim = 10)				RaTSLPSO (Dim = 10)			
	Min	Max	Mean	Std	Min	Max	Mean	Std
F1	1.31 × 10^2^	**1.43 × 10^3^**	7.12 × 10^2^	**3.85 × 10^2^**	**1.00 × 10^2^**	1.46 × 10^3^	**6.30 × 10^2^**	4.44 × 10^2^
F3	3.00 × 10^2^	**3.00 × 10^2^**	**3.00 × 10^2^**	**1.26 × 10^−5^**	3.00 × 10^2^	3.00 × 10^2^	3.00 × 10^2^	2.22 × 10^−3^
F4	4.00 × 10^2^	4.05 × 10^2^	**4.01 × 10^2^**	**1.40 × 10^0^**	**4.00 × 10^2^**	**4.05 × 10^2^**	4.02 × 10^2^	1.76 × 10^0^
F5	5.03 × 10^2^	5.14 × 10^2^	5.08 × 10^2^	2.56 × 10^0^	5.03 × 10^2^	**5.10 × 10^2^**	**5.07 × 10^2^**	**1.74 × 10^0^**
F6	6.00 × 10^2^	6.00 × 10^2^	6.00 × 10^2^	2.72 × 10^−5^	6.00 × 10^2^	**6.00 × 10^2^**	**6.00 × 10^2^**	**3.44 × 10^−9^**
F7	7.12 × 10^2^	**7.23 × 10^2^**	**7.19 × 10^2^**	**2.82 × 10^0^**	**7.08 × 10^2^**	7.23 × 10^2^	7.19 × 10^2^	3.43 × 10^0^
F8	8.04 × 10^2^	8.12 × 10^2^	8.08 × 10^2^	2.25 × 10^0^	**8.03 × 10^2^**	**8.10 × 10^2^**	**8.07 × 10^2^**	**1.79 × 10^0^**
F9	9.00 × 10^2^	9.02 × 10^2^	9.00 × 10^2^	4.43 × 10^−1^	9.00 × 10^2^	**9.00 × 10^2^**	**9.00 × 10^2^**	**2.73 × 10^−2^**
F10	1.01 × 10^3^	1.40 × 10^3^	1.21 × 10^3^	**1.05 × 10^2^**	**1.00 × 10^3^**	**1.36 × 10^3^**	**1.21 × 10^3^**	1.16 × 10^2^
F11	**1.10 × 10^3^**	1.11 × 10^3^	1.10 × 10^3^	1.94 × 10^0^	1.10 × 10^3^	**1.11 × 10^3^**	**1.10 × 10^3^**	**1.38 × 10^0^**
F12	2.41 × 10^3^	2.98 × 10^4^	1.42 × 10^4^	8.42 × 10^3^	**2.13 × 10^3^**	**2.25 × 10^4^**	**1.13 × 10^4^**	**6.36 × 10^3^**
F13	**1.31 × 10^3^**	**1.46 × 10^3^**	**1.37 × 10^3^**	**3.47 × 10^1^**	1.31 × 10^3^	1.98 × 10^3^	1.52 × 10^3^	1.99 × 10^2^
F14	1.42 × 10^3^	1.48 × 10^3^	1.44 × 10^3^	1.27 × 10^1^	**1.41 × 10^3^**	**1.44 × 10^3^**	**1.43 × 10^3^**	**6.42 × 10^0^**
F15	1.50 × 10^3^	1.58 × 10^3^	1.53 × 10^3^	1.91 × 10^1^	**1.50 × 10^3^**	**1.53 × 10^3^**	**1.52 × 10^3^**	**8.45 × 10^0^**
F16	**1.60 × 10^3^**	1.62 × 10^3^	1.61 × 10^3^	4.05 × 10^0^	1.60 × 10^3^	**1.61 × 10^3^**	**1.60 × 10^3^**	**1.85 × 10^0^**
F17	1.71 × 10^3^	1.75 × 10^3^	1.73 × 10^3^	7.73 × 10^0^	**1.71 × 10^3^**	**1.74 × 10^3^**	**1.73 × 10^3^**	**6.45 × 10^0^**
F18	**1.85 × 10^3^**	3.34 × 10^3^	**2.27 × 10^3^**	2.99 × 10^2^	1.96 × 10^3^	**2.87 × 10^3^**	2.38 × 10^3^	**2.88 × 10^2^**
F19	**1.90 × 10^3^**	1.92 × 10^3^	1.91 × 10^3^	3.61 × 10^0^	1.90 × 10^3^	**1.91 × 10^3^**	**1.91 × 10^3^**	**2.47 × 10^0^**
F20	2.00 × 10^3^	2.04 × 10^3^	2.02 × 10^3^	**6.89 × 10^0^**	**2.00 × 10^3^**	2.03 × 10^3^	2.02 × 10^3^	9.97 × 10^0^
F21	2.20 × 10^3^	**2.20 × 10^3^**	2.20 × 10^3^	**7.51 × 10^−13^**	2.20 × 10^3^	2.20 × 10^3^	2.20 × 10^3^	2.14 × 10^−7^
F22	2.20 × 10^3^	2.30 × 10^3^	2.23 × 10^3^	3.81 × 10^1^	2.20 × 10^3^	2.30 × 10^3^	**2.23 × 10^3^**	**3.80 × 10^1^**
F23	2.65 × 10^3^	2.67 × 10^3^	2.66 × 10^3^	**3.51 × 10^0^**	**2.64 × 10^3^**	**2.66 × 10^3^**	**2.66 × 10^3^**	5.08 × 10^0^
F24	2.43 × 10^3^	2.66 × 10^3^	2.53 × 10^3^	4.81 × 10^1^	**2.40 × 10^3^**	**2.54 × 10^3^**	**2.50 × 10^3^**	**2.95 × 10^1^**
F25	**2.85 × 10^3^**	2.90 × 10^3^	2.90 × 10^3^	9.15 × 10^0^	2.85 × 10^3^	**2.90 × 10^3^**	**2.90 × 10^3^**	**8.92 × 10^0^**
F26	**2.60 × 10^3^**	2.95 × 10^3^	2.74 × 10^3^	1.45 × 10^2^	2.60 × 10^3^	**2.90 × 10^3^**	**2.73 × 10^3^**	**1.38 × 10^2^**
F27	3.11 × 10^3^	3.15 × 10^3^	3.13 × 10^3^	**1.16 × 10^1^**	**3.10 × 10^3^**	**3.15 × 10^3^**	**3.13 × 10^3^**	1.42 × 10^1^
F28	3.10 × 10^3^	3.15 × 10^3^	3.13 × 10^3^	**2.22 × 10^1^**	**3.06 × 10^3^**	**3.15 × 10^3^**	**3.12 × 10^3^**	2.44 × 10^1^
F29	3.14 × 10^3^	3.19 × 10^3^	3.16 × 10^3^	**1.25 × 10^1^**	**3.10 × 10^3^**	**3.17 × 10^3^**	**3.15 × 10^3^**	1.66 × 10^1^
F30	**3.43 × 10^3^**	2.15 × 10^4^	7.91 × 10^3^	4.06 × 10^3^	3.75 × 10^3^	**1.04 × 10^4^**	**7.26 × 10^3^**	**2.13 × 10^3^**

**Table 7 entropy-23-01200-t007:** Comparison of experimental results of TSLPSO and RaTSLPSO in the case of 30 dimensions.

Fun	TSLPSO (Dim = 30)				RaTSLPSO (Dim = 30)			
	Min	Max	Mean	Std	Min	Max	Mean	Std
F1	**1.01 × 10^2^**	4.46 × 10^3^	1.39 × 10^3^	1.21 × 10^3^	1.03 × 10^2^	**2.49 × 10^3^**	**1.11 × 10^3^**	**9.16 × 10^2^**
F3	3.46 × 10^2^	1.07 × 10^4^	1.78 × 10^3^	2.08 × 10^3^	**3.31 × 10^2^**	**7.38 × 10^2^**	**4.37 × 10^2^**	**1.36 × 10^2^**
F4	4.00 × 10^2^	5.44 × 10^2^	4.47 × 10^2^	4.22 × 10^1^	**4.00 × 10^2^**	**4.70 × 10^2^**	**4.22 × 10^2^**	**3.00 × 10^1^**
F5	**5.36 × 10^2^**	6.03 × 10^2^	5.63 × 10^2^	1.54 × 10^1^	5.38 × 10^2^	**5.62 × 10^2^**	**5.53 × 10^2^**	**7.30 × 10^0^**
F6	6.00 × 10^2^	6.00 × 10^2^	6.00 × 10^2^	2.01 × 10^−7^	6.00 × 10^2^	**6.00 × 10^2^**	6.00 × 10^2^	**2.10 × 10^−13^**
F7	7.70 × 10^2^	8.41 × 10^2^	7.94 × 10^2^	1.41 × 10^1^	**7.61 × 10^2^**	**7.94 × 10^2^**	**7.84 × 10^2^**	**8.10 × 10^0^**
F8	8.36 × 10^2^	9.03 × 10^2^	8.64 × 10^2^	1.55 × 10^1^	8.36 × 10^2^	**8.62 × 10^2^**	**8.51 × 10^2^**	**7.44 × 10^0^**
F9	9.01 × 10^2^	2.09 × 10^3^	1.05 × 10^3^	2.46 × 10^2^	**9.01 × 10^2^**	**9.95 × 10^2^**	**9.45 × 10^2^**	**3.26 × 10^1^**
F10	**1.97 × 10^3^**	4.42 × 10^3^	3.23 × 10^3^	6.23 × 10^2^	2.29 × 10^3^	**3.32 × 10^3^**	**2.85 × 10^3^**	**2.88 × 10^2^**
F11	**1.12 × 10^3^**	1.30 × 10^3^	1.18 × 10^3^	4.69 × 10^1^	1.13 × 10^3^	**1.16 × 10^3^**	**1.15 × 10^3^**	**1.20 × 10^1^**
F12	**2.45 × 10^3^**	1.86 × 10^4^	7.36 × 10^3^	4.16 × 10^3^	2.78 × 10^3^	**6.16 × 10^3^**	**4.35 × 10^3^**	**1.03 × 10^3^**
F13	**1.34 × 10^3^**	2.12 × 10^3^	1.66 × 10^3^	2.27 × 10^2^	1.34 × 10^3^	**1.88 × 10^3^**	**1.57 × 10^3^**	**1.66 × 10^2^**
F14	**1.43 × 10^3^**	1.21 × 10^4^	2.21 × 10^3^	2.01 × 10^3^	1.44 × 10^3^	**1.47 × 10^3^**	**1.45 × 10^3^**	**9.13 × 10^0^**
F15	**1.52 × 10^3^**	**1.78 × 10^3^**	**1.65 × 10^3^**	**8.34 × 10^1^**	1.56 × 10^3^	2.57 × 10^3^	1.81 × 10^3^	2.74 × 10^2^
F16	**1.78 × 10^3^**	2.35 × 10^3^	2.04 × 10^3^	1.19 × 10^2^	1.83 × 10^3^	**1.99 × 10^3^**	**1.92 × 10^3^**	**4.91 × 10^1^**
F17	**1.77 × 10^3^**	2.07 × 10^3^	1.92 × 10^3^	8.16 × 10^1^	1.78 × 10^3^	**1.92 × 10^3^**	**1.87 × 10^3^**	**3.70 × 10^1^**
F18	**1.29 × 10^4^**	2.43 × 10^5^	6.93 × 10^4^	6.14 × 10^4^	1.59 × 10^4^	**5.67 × 10^4^**	**3.76 × 10^4^**	**1.06 × 10^4^**
F19	**1.92 × 10^3^**	**3.28 × 10^3^**	**2.26 × 10^3^**	**3.32 × 10^2^**	1.94 × 10^3^	4.84 × 10^3^	2.71 × 10^3^	8.30 × 10^2^
F20	**2.08 × 10^3^**	2.30 × 10^3^	2.21 × 10^3^	5.27 × 10^1^	2.09 × 10^3^	**2.21 × 10^3^**	**2.15 × 10^3^**	**3.64 × 10^1^**
F21	2.11 × 10^3^	2.23 × 10^3^	2.17 × 10^3^	**2.61 × 10^1^**	**2.10 × 10^3^**	**2.17 × 10^3^**	**2.15 × 10^3^**	3.44 × 10^1^
F22	2.23 × 10^3^	2.30 × 10^3^	2.27 × 10^3^	1.70 × 10^1^	**2.23 × 10^3^**	**2.27 × 10^3^**	**2.25 × 10^3^**	**1.14 × 10^1^**
F23	**2.82 × 10^3^**	2.90 × 10^3^	2.85 × 10^3^	1.89 × 10^1^	2.83 × 10^3^	**2.86 × 10^3^**	**2.85 × 10^3^**	**1.04 × 10^1^**
F24	2.60 × 10^3^	3.41 × 10^3^	3.29 × 10^3^	**2.34 × 10^2^**	2.60 × 10^3^	**3.35 × 10^3^**	**2.85 × 10^3^**	2.78 × 10^2^
F25	**2.90 × 10^3^**	2.99 × 10^3^	2.93 × 10^3^	2.92 × 10^1^	2.90 × 10^3^	**2.92 × 10^3^**	**2.92 × 10^3^**	**4.89 × 10^0^**
F26	2.90 × 10^3^	5.24 × 10^3^	4.17 × 10^3^	8.37 × 10^2^	2.90 × 10^3^	**4.50 × 10^3^**	**3.43 × 10^3^**	**5.46 × 10^2^**
F27	**3.45 × 10^3^**	3.56 × 10^3^	**3.49 × 10^3^**	2.65 × 10^1^	3.46 × 10^3^	**3.54 × 10^3^**	3.51 × 10^3^	**2.26 × 10^1^**
F28	3.22 × 10^3^	3.44 × 10^3^	3.26 × 10^3^	5.64 × 10^1^	**3.18 × 10^3^**	**3.28 × 10^3^**	**3.23 × 10^3^**	**2.88 × 10^1^**
F29	3.32 × 10^3^	3.61 × 10^3^	3.45 × 10^3^	7.47 × 10^1^	**3.26 × 10^3^**	**3.43 × 10^3^**	**3.38 × 10^3^**	**5.17 × 10^1^**
F30	4.13 × 10^3^	**1.96 × 10^4^**	**9.30 × 10^3^**	**4.27 × 10^3^**	**4.11 × 10^3^**	3.45 × 10^4^	1.61 × 10^4^	9.66 × 10^3^

**Table 8 entropy-23-01200-t008:** Comparison of experimental results of HCLPSO and RaHCLPSO in the case of 10 dimensions.

Fun	HCLPSO (Dim = 10)				RaHCLPSO (Dim = 10)			
	Min	Max	Mean	Std	Min	Max	Mean	Std
F1	1.07 × 10^2^	1.53 × 10^3^	4.30 × 10^2^	4.29 × 10^2^	**1.00 × 10^2^**	**3.48 × 10^2^**	**1.75 × 10^2^**	**8.54 × 10^1^**
F3	**3.00 × 10^2^**	**3.00 × 10^2^**	**3.00 × 10^2^**	**1.03 × 10^−8^**	3.00 × 10^2^	3.00 × 10^2^	3.00 × 10^2^	**5.51 × 10^−3^**
F4	4.00 × 10^2^	4.05 × 10^2^	4.01 × 10^2^	1.55 × 10^0^	**4.00 × 10^2^**	**4.00 × 10^2^**	**4.00 × 10^2^**	**1.01 × 10^−1^**
F5	5.02 × 10^2^	5.12 × 10^2^	5.06 × 10^2^	2.29 × 10^0^	**5.02 × 10^2^**	**5.05 × 10^2^**	**5.05 × 10^2^**	**8.56 × 10^−1^**
F6	**6.00 × 10^2^**	**6.00 × 10^2^**	**6.00 × 10^2^**	**7.44 × 10^−8^**	6.00 × 10^2^	6.00 × 10^2^	6.00 × 10^2^	**1.15 × 10^−3^**
F7	7.14 × 10^2^	7.29 × 10^2^	7.18 × 10^2^	3.50 × 10^0^	**7.13 × 10^2^**	**7.18 × 10^2^**	**7.16 × 10^2^**	**1.50 × 10^0^**
F8	**8.02 × 10^2^**	8.09 × 10^2^	8.06 × 10^2^	1.97 × 10^0^	8.02 × 10^2^	**8.06 × 10^2^**	**8.05 × 10^2^**	**1.18 × 10^0^**
F9	**9.00 × 10^2^**	**9.00 × 10^2^**	**9.00 × 10^2^**	**1.64 × 10^−13^**	9.00 × 10^2^	9.00 × 10^2^	9.00 × 10^2^	4.36 × 10^−5^
F10	1.01 × 10^3^	1.38 × 10^3^	1.14 × 10^3^	1.16 × 10^2^	**1.00 × 10^3^**	**1.14 × 10^3^**	**1.05 × 10^3^**	**4.50 × 10^1^**
F11	**1.10 × 10^3^**	1.10 × 10^3^	1.10 × 10^3^	1.07 × 10^0^	1.10 × 10^3^	**1.10 × 10^3^**	**1.10 × 10^3^**	**7.65 × 10^−1^**
F12	2.61 × 10^3^	4.38 × 10^4^	1.30 × 10^4^	9.90 × 10^3^	**2.57 × 10^3^**	**2.02 × 10^4^**	**1.07 × 10^4^**	**5.24 × 10^3^**
F13	**1.30 × 10^3^**	**1.42 × 10^3^**	**1.35 × 10^3^**	**3.24 × 10^1^**	1.31 × 10^3^	1.49 × 10^3^	1.39 × 10^3^	5.37 × 10^1^
F14	**1.42 × 10^3^**	1.47 × 10^3^	1.45 × 10^3^	1.21 × 10^1^	1.43 × 10^3^	**1.45 × 10^3^**	**1.44 × 10^3^**	**7.23 × 10^0^**
F15	**1.51 × 10^3^**	1.56 × 10^3^	1.52 × 10^3^	1.13 × 10^1^	1.51 × 10^3^	**1.53 × 10^3^**	**1.52 × 10^3^**	**7.12 × 10^0^**
F16	1.60 × 10^3^	1.62 × 10^3^	1.60 × 10^3^	3.28 × 10^0^	**1.60 × 10^3^**	**1.60 × 10^3^**	**1.60 × 10^3^**	**6.90 × 10^−1^**
F17	1.72 × 10^3^	1.74 × 10^3^	1.73 × 10^3^	**4.61 × 10^0^**	**1.70 × 10^3^**	**1.73 × 10^3^**	**1.73 × 10^3^**	6.16 × 10^0^
F18	1.93 × 10^3^	4.12 × 10^3^	2.62 × 10^3^	5.66 × 10^2^	**1.89 × 10^3^**	**2.69 × 10^3^**	**2.31 × 10^3^**	**2.63 × 10^2^**
F19	**1.90 × 10^3^**	1.94 × 10^3^	**1.91 × 10^3^**	8.08 × 10^0^	1.90 × 10^3^	**1.91 × 10^3^**	1.91 × 10^3^	**2.44 × 10^0^**
F20	2.00 × 10^3^	2.04 × 10^3^	2.01 × 10^3^	1.12 × 10^1^	**2.00 × 10^3^**	**2.02 × 10^3^**	**2.01 × 10^3^**	**9.24 × 10^0^**
F21	2.20 × 10^3^	**2.20 × 10^3^**	**2.20 × 10^3^**	**3.97 × 10^−6^**	**2.10 × 10^3^**	2.26 × 10^3^	2.23 × 10^3^	4.20 × 10^1^
F22	**2.20 × 10^3^**	**2.30 × 10^3^**	2.27 × 10^3^	**4.35 × 10^1^**	2.20 × 10^3^	2.39 × 10^3^	**2.25 × 10^3^**	7.68 × 10^1^
F23	**2.40 × 10^3^**	2.66 × 10^3^	2.64 × 10^3^	4.71 × 10^1^	2.55 × 10^3^	**2.65 × 10^3^**	**2.63 × 10^3^**	**2.81 × 10^1^**
F24	2.50 × 10^3^	2.55 × 10^3^	2.50 × 10^3^	1.20 × 10^1^	**2.49 × 10^3^**	**2.50 × 10^3^**	**2.50 × 10^3^**	**2.34 × 10^0^**
F25	2.89 × 10^3^	2.90 × 10^3^	2.90 × 10^3^	1.14 × 10^0^	**2.89 × 10^3^**	**2.90 × 10^3^**	**2.90 × 10^3^**	**7.35 × 10^−1^**
F26	**2.60 × 10^3^**	2.90 × 10^3^	2.73 × 10^3^	1.25 × 10^2^	2.60 × 10^3^	**2.82 × 10^3^**	**2.66 × 10^3^**	**9.55 × 10^1^**
F27	3.10 × 10^3^	3.14 × 10^3^	3.11 × 10^3^	1.34 × 10^1^	3.10 × 10^3^	**3.11 × 10^3^**	**3.10 × 10^3^**	**4.33 × 10^0^**
F28	**3.08 × 10^3^**	3.15 × 10^3^	3.10 × 10^3^	1.33 × 10^1^	3.08 × 10^3^	**3.10 × 10^3^**	**3.10 × 10^3^**	**2.75 × 10^0^**
F29	3.13 × 10^3^	3.18 × 10^3^	3.15 × 10^3^	1.08 × 10^1^	**3.12 × 10^3^**	**3.15 × 10^3^**	**3.14 × 10^3^**	**5.79 × 10^0^**
F30	**3.41 × 10^3^**	1.17 × 10^4^	5.38 × 10^3^	2.31 × 10^3^	3.67 × 10^3^	**5.66 × 10^3^**	**4.58 × 10^3^**	**5.92 × 10^2^**

**Table 9 entropy-23-01200-t009:** Comparison of experimental results of HCLPSO and RaHCLPSO in the case of 30 dimensions.

Fun	HCLPSO (Dim = 30)				RaHCLPSO (Dim = 30)			
	Min	Max	Mean	Std	Min	Max	Mean	Std
F1	**1.00 × 10^2^**	1.32 × 10^3^	4.10 × 10^2^	3.47 × 10^2^	1.08 × 10^2^	**8.05 × 10^2^**	**3.47 × 10^2^**	**2.53 × 10^2^**
F3	**3.00 × 10^2^**	3.05 × 10^2^	3.01 × 10^2^	1.35 × 10^0^	3.01 × 10^2^	**3.01 × 10^2^**	**3.01 × 10^2^**	**1.52 × 10^−1^**
F4	4.04 × 10^2^	5.15 × 10^2^	4.74 × 10^2^	**2.32 × 10^1^**	**4.03 × 10^2^**	**4.72 × 10^2^**	**4.52 × 10^2^**	2.89 × 10^1^
F5	5.19 × 10^2^	5.58 × 10^2^	5.37 × 10^2^	9.26 × 10^0^	**5.17 × 10^2^**	**5.36 × 10^2^**	**5.29 × 10^2^**	**6.47 × 10^0^**
F6	**6.00 × 10^2^**	**6.00 × 10^2^**	**6.00 × 10^2^**	**2.02 × 10^−6^**	6.00 × 10^2^	6.00 × 10^2^	6.00 × 10^2^	8.97 × 10^−4^
F7	**7.52 × 10^2^**	8.16 × 10^2^	7.83 × 10^2^	1.70 × 10^1^	7.61 × 10^2^	**7.90 × 10^2^**	**7.79 × 10^2^**	**9.24 × 10^0^**
F8	**8.21 × 10^2^**	8.65 × 10^2^	8.40 × 10^2^	1.21 × 10^1^	8.25 × 10^2^	**8.38 × 10^2^**	**8.32 × 10^2^**	**3.31 × 10^0^**
F9	9.00 × 10^2^	9.15 × 10^2^	9.03 × 10^2^	4.02 × 10^0^	**9.00 × 10^2^**	**9.02 × 10^2^**	**9.01 × 10^2^**	**5.40 × 10^−1^**
F10	2.06 × 10^3^	3.92 × 10^3^	2.91 × 10^3^	4.45 × 10^2^	**1.96 × 10^3^**	**2.65 × 10^3^**	**2.43 × 10^3^**	**1.87 × 10^2^**
F11	**1.13 × 10^3^**	1.24 × 10^3^	1.15 × 10^3^	2.51 × 10^1^	1.13 × 10^3^	**1.15 × 10^3^**	**1.14 × 10^3^**	**8.40 × 10^0^**
F12	**2.17 × 10^3^**	**1.05 × 10^4^**	**4.62 × 10^3^**	**1.97 × 10^3^**	8.21 × 10^3^	2.72 × 10^4^	1.96 × 10^4^	5.33 × 10^3^
F13	**1.33 × 10^3^**	**1.60 × 10^3^**	**1.43 × 10^3^**	**7.55 × 10^1^**	1.36 × 10^3^	1.83 × 10^3^	1.49 × 10^3^	1.50 × 10^2^
F14	**1.45 × 10^3^**	**1.60 × 10^3^**	**1.52 × 10^3^**	**3.89 × 10^1^**	1.47 × 10^3^	1.59 × 10^3^	1.53 × 10^3^	4.29 × 10^1^
F15	**1.53 × 10^3^**	**1.79 × 10^3^**	**1.62 × 10^3^**	**7.42 × 10^1^**	1.55 × 10^3^	2.34 × 10^3^	1.94 × 10^3^	2.62 × 10^2^
F16	**1.61 × 10^3^**	2.32 × 10^3^	1.94 × 10^3^	1.74 × 10^2^	1.62 × 10^3^	**1.96 × 10^3^**	**1.82 × 10^3^**	**1.09 × 10^2^**
F17	1.77 × 10^3^	1.94 × 10^3^	1.83 × 10^3^	4.61 × 10^1^	**1.77 × 10^3^**	**1.82 × 10^3^**	**1.80 × 10^3^**	**1.20 × 10^1^**
F18	**7.96 × 10^3^**	1.13 × 10^5^	3.49 × 10^4^	2.69 × 10^4^	1.14 × 10^4^	**3.66 × 10^4^**	**2.68 × 10^4^**	**6.88 × 10^3^**
F19	**1.93 × 10^3^**	**2.13 × 10^3^**	**2.01 × 10^3^**	**5.89 × 10^1^**	1.94 × 10^3^	2.54 × 10^3^	2.10 × 10^3^	2.02 × 10^2^
F20	**2.06 × 10^3^**	2.27 × 10^3^	2.15 × 10^3^	6.79 × 10^1^	2.06 × 10^3^	**2.14 × 10^3^**	**2.10 × 10^3^**	**2.43 × 10^1^**
F21	2.25 × 10^3^	2.25 × 10^3^	2.25 × 10^3^	**1.08 × 10^−12^**	**2.12 × 10^3^**	**2.18 × 10^3^**	**2.17 × 10^3^**	1.98 × 10^1^
F22	2.35 × 10^3^	2.35 × 10^3^	2.35 × 10^3^	**1.33 × 10^−12^**	**2.22 × 10^3^**	**2.24 × 10^3^**	**2.23 × 10^3^**	3.85 × 10^0^
F23	2.82 × 10^3^	2.88 × 10^3^	2.85 × 10^3^	1.43 × 10^1^	**2.78 × 10^3^**	**2.84 × 10^3^**	**2.83 × 10^3^**	**1.71 × 10^1^**
F24	**2.60 × 10^3^**	**2.60 × 10^3^**	**2.60 × 10^3^**	**5.28 × 10^−7^**	2.60 × 10^3^	2.61 × 10^3^	2.60 × 10^3^	**9.18 × 10^−1^**
F25	**2.90 × 10^3^**	2.97 × 10^3^	2.91 × 10^3^	1.72 × 10^1^	2.90 × 10^3^	**2.92 × 10^3^**	**2.91 × 10^3^**	**7.42 × 10^0^**
F26	**2.90 × 10^3^**	2.90 × 10^3^	**2.90 × 10^3^**	**3.71 × 10^−3^**	2.90 × 10^3^	**2.90 × 10^3^**	2.90 × 10^3^	**4.74 × 10^−3^**
F27	3.39 × 10^3^	3.63 × 10^3^	3.54 × 10^3^	6.41 × 10^1^	**3.35 × 10^3^**	**3.51 × 10^3^**	**3.46 × 10^3^**	**4.44 × 10^1^**
F28	3.14 × 10^3^	3.30 × 10^3^	3.23 × 10^3^	3.42 × 10^1^	**3.14 × 10^3^**	**3.22 × 10^3^**	**3.19 × 10^3^**	**2.94 × 10^1^**
F29	**3.21 × 10^3^**	3.41 × 10^3^	3.29 × 10^3^	5.57 × 10^1^	3.26 × 10^3^	**3.30 × 10^3^**	**3.28 × 10^3^**	**1.33 × 10^1^**
F30	**4.22 × 10^3^**	1.56 × 10^4^	**7.68 × 10^3^**	**3.06 × 10^3^**	9.50 × 10^3^	**3.57 × 10^4^**	2.69 × 10^4^	7.65 × 10^3^

## Data Availability

Not applicable.

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
