# Peer review of "A Particle Swarm Algorithm Based on a Multi-Stage Search Strategy"

_entropy, 2021, doi:10.3390/e23091200_

Round 1

Reviewer 1 Report

In this article, the authors discuss a relevant topic within bioinspired computing. However, in general, the text is poorly written and disorganized. Several problems make it difficult to understand and read the content, some of which I will list below:

  • The text needs grammar proofing, preferably by an English language expert; it is also necessary to review the material considering it is a scientific work. There is a specific style to be followed, and that, here, is being misused.
  • There are several equations in the manuscript, but the authors do not detail all their components.
  • Figures are not self-explained, and their captions are insufficient to guide the reader. Besides, the authors do not refer to several images and tables in the text.
  • The manuscript lacks details about the experimental design. Moreover, there is no explanation about how the authors obtained the parameters showed in Table 2.

The paper also has some conceptual mistakes. For instance, the authors state that the "particle swarm algorithm finds the optimal position," but this is not true. Metaheuristics tend to a global convergence, but there are no guarantees over the solution quality.

Finally, the proposed approach requires several parameters; so, I can't entirely agree with the following statement: "we proposed a more direct and simple way to improve the particle swarm."

Reviewer 2 Report

"The particle swarm algorithm has the disadvantages of early maturity, easily falling into local optima, and low search accuracy."
Instead of early maturity (which does not make sense in the algorithmic analogue), premature convergence is the proper phrase. Also, premature convergence is the same as trapping (not falling) in local optima.

Also, what "... has been used to improve the diversity, search accuracy, and results of many algorithms"? This is not clear.

pbest and gbest are not "extreme values", but the best position found by the corresponding particle and the best position found by the whole population up to the current iteration of the algorithm, respectively.

Therefore, the position of each particle is not the one described in line 107.

The statement "TSLPSO is currently one of the algorithms that have achieved better results on the congress on evolutionary computation(CEC) test function" should be supported by a relevant citation.

Premature convergence is not observed only when most of the particles are gathered with close range of the optimum. Usually, the actual optimum cannot be found in a reasonable amount of time. Premature convergence occurs every time that most of the particles get near a local optimum and they are trapped there because PSO movement does not let them perform wide jumps.

All notations that occur in every equation should be explained. For example, pbestdfl(d) is not clear what represents.

Fig. 4 should be enlarged, so that all details are visible.

The analysis of results could be extended, in order to give more insight on results and discuss if the initial research hypothesis has been confirmed.

Minor comment:
In lots of cases, first person is observed (i.e. "I", "we"). Although, "we" may be accepted (while it is preferred to write in third person), "I" should me avoided.

Round 2

Reviewer 1 Report

The authors adapted the article concerning my comments.

This manuscript is a resubmission of an earlier submission. The following is a list of the peer review reports and author responses from that submission.